# Leveraging Label Dependencies for Calibration in Multi-Label Classification through Proper Scoring Rule

## Abstract

Modern Deep Neural Networks (DNNs) trained by using cross entropy for binary or multi-class classification are known to produce poorly calibrated probability estimates. While various calibration methods have been proposed, only a few addresses the challenge of calibrating Multi-Label Classification (MLC) tasks. Multi-label classification is essential in real-world applications, as most objects or instances naturally belong to multiple categories, and the associated labels often exhibit strong interdependencies. A key difficulty in calibrating MLC models lies in effectively considering the information of label interdependencies. Existing methods that attempt to model the label interdependencies often lack rigorous statistical justification or they consider the labels are independent or lacks being strictly proper - a property which induces calibrated predicted probabilities upon minimization. In this work, we introduce a novel loss function, *Correlated Multi-Label Loss (CMLL)*, that explicitly captures label interdependencies while satisfying the properties of a strictly proper loss. Our method leverages pairwise label correlations to incorporate dependency information into the training process and is proven to be Fisher consistent. Extensive experiments on three publicly available benchmark multi-label datasets demonstrate the effectiveness of our approach. Our proposed method significantly reduces calibration error while maintaining state-of-the-art classification accuracy.

## 1 Introduction

Modern Deep Neural Networks (DNNs) including Convolutional Neural Networks (CNNs) (Li et al., 2021) and Vision Transformers (ViTs) (Dosovitskiy et al., 2020) have demonstrated superior capacity in supervised learning tasks in computer vision. However, application of DNNs in safety critical tasks requires trustworthy ML models which are not only accurate but also predicts accurate estimate of posterior probabilities i.e. the models should be well calibrated. For example, A perfectly calibrated weather prediction model would predict an $80\%$ chance of rain on multiple days throughout the year, and it would actually rain on exactly $80\%$ of those days. The importance of model calibration has been highlighted in several tasks. In disease diagnosis (Mehrtash et al., 2020; Gawlikowski et al., 2023), the predicted probabilities (confidence) of a model can be used to determine whether human intervention is necessary. But this can only be used safely if the DNN is well calibrated. Application of DNNs in cost-sensitive areas such as decision making in business (Petrides et al., 2022; Manzoor et al., 2024), fraud detection (Bahnsen et al., 2013) etc, involves different costs for different misclassification errors. In these cases, building a model that predicts accurate class posterior probabilities cannot be compromised in favour of achieving Bayes optimality. Hence, calibration of DNNs has attracted substantial attention in several areas of computer vision and machine learning tasks (Gal & Ghahramani, 2016; Guo et al., 2017; Lakshminarayanan et al., 2017; Bouniot et al., 2023).

While various calibration methods have been proposed, most of them address binary or multi-class classification problems rather than MLC problems. However, in real world situations multiple labels are often associated with a single instance. For instance, in autonomous driving, a single image is often annotated with multiple scene instances (Chen et al., 2019) or in medical images like Chest X-Ray (Baltruschat et al., 2019) where multiple disease condition might be

associated with the same chest X-ray image. Thus, multi-label learning has garnered significant attention, with a focus on enhancing both accuracy and trustworthiness, particularly due to its applications in safety-critical domains (Reiß et al., 2021; Baltruschat et al., 2019). Some of the most popular approaches to multi-label learning include designing novel loss functions (Ridnik et al., 2021; Gong et al., 2013; Kobayashi, 2023; Wu et al., 2020) and leveraging auxiliary information (Yazici et al., 2020; Guo et al., 2023; Ding et al., 2023). For MLC tasks, DNNs can be trained using Class Probability Estimate (CPE) losses like Binary Cross Entropy (BCE) which encourages probability calibration, but they do not account for the information of label interdependency. Multi-Label datasets can show complex interdependencies (For instance, a cow is more likely to appear in the vicinity of grass than in the vicinity of a computer) among the labels which if not modelled properly may lead to problems in predicting accurate posterior probability estimates that is used in further downstream tasks like decision making and uncertainty quantification. A series of studies (Liu & Tsang, 2015b; Shen et al., 2017; Liu & Tsang, 2015a; Chen & Lin, 2012) have highlighted that the assumption of independence of labels can result in degraded performance. There are a few methods (Chen & Lin, 2012; Andrew et al., 2013; Read et al., 2011) that tries to model the dependency among the label during classification but the question of whether these or similar losses also encourage to produce accurate probability estimate is still open. Besides, statistical properties of these methods are still not well understood. In our work we aim to bridge this gap by proposing a *Proper Scoring Rule* (PSR) to train MLC tasks.

In our studies we have found that multi-label DNNs trained with conventional loss functions tend to produce poorly calibrated probability estimates. We argue that this arises because conventional loss functions are either not Strictly Proper (Gneiting & Raftery, 2007) or they fail to capture the dependencies among labels. Strictly Proper Scoring Rules (PSR) are a class of loss functions which gets minimized uniquely by true posterior probability estimates. In other words models trained with Strictly PSRs should return accurate posterior probabilities thus making the model highly calibrated. Classical loss functions like BCE is a PSR but it cannot handle the information of label interdependencies; asymmetric loss functions like Focal Loss (Mukhoti et al., 2020) are not Strictly PSR. Thus, conventional loss functions cannot ensure the calibration of a DNN, as most of them fail to satisfy both conditions.

Motivated by the need of a loss function for MLC tasks which is Strictly PSR and also considers label dependency, we propose Correlated Multi-Label Loss (CMLL) which satisfies all these constraints. This novel loss function captures difference between correlation of logits of two labels and correlation of ground truths of the same two labels. Experiments on several multi-label datasets shows the superiority of our loss function in terms of calibration while keeping the accuracy on par with the other popularly used loss functions. The main contributions of our work are as follows:

- We propose a novel loss function for MLC tasks that captures dependency among labels.
- Theoretical analysis is provided to establish the soundness of the proposed loss.
- The loss function is shown to constitute a Proper Scoring Rule, a property that is necessary for providing reliable probability estimates.
- We also provide generalization analysis of our proposed loss function and found linear dependency on the number of labels.
- Extensive experiments on three publicly available benchmark multi-label datasets establish that our loss function produces better calibrated results for MLC tasks than existing loss functions.

## 2 RELATED WORKS

**Calibration:** Several studies in recent years has revealed that modern neural networks are poorly calibrated i.e. over confident (Guo et al., 2017; Tao et al., 2023). Various methods have been proposed for calibration including post hoc methods such as Temperature Scaling (Guo et al., 2017), Histogram Binning (Zadrozny & Elkan, 2001), Dirichlet Calibration (Kull et al., 2019); train-time regularization (Popordanoska et al., 2022); Bayesian Methods (Naeini et al., 2015). Results on these methods are available only for binary and multi-class problems, not for multi-label classification

problems. For calibration in MLC tasks Cheng & Vasconcelos (2024); Ridnik et al. (2021) has suggested Asymmetric Proper Loss but their method assumes label independence. Pal et al. (2025) has suggested considering pairwise predicted probabilities in the training process but their loss function is not strictly proper. Peng et al. (2023) has suggested semantic aware regularization to incorporate the information of dependency in sequential confidence calibration but their work is not designed to handle MLC tasks. Prior work has attempted to incorporate label dependency information into multi-label learning. In particular, Chen et al. (2024) leverage category-specific features to adapt label smoothing (Müller et al., 2019) for multi-label classification. However, the proposed DCLR framework in Chen et al. (2024) relies on a two-stage learning procedure, which increases the overall training complexity. Moreover, the resulting objective does not constitute a proper scoring rule. Also, the method of Chen et al. (2024) is restricted to a limited class of architectures.

**Proper Class Probability Estimation (CPE) Losses:** Proper CPE losses are designed to be minimized by accurate probability estimates. These class of loss functions have long been studied in statistics (Hendrickson & Buehler, 1971) and later in machine learning (Gneiting & Raftery, 2007; Cheng & Vasconcelos, 2024). For MLC tasks, Cheng & Vasconcelos (2024) has shown that Asymmetric Proper losses produces better calibrated results but they considered the labels are independent. To the best of our knowledge, strictly proper class probability estimates has not yet been studied previously in MLC tasks with label dependency.

**Multi-Label Classification:** Multi-label tasks, which allow multiple labels to be present in a single instance simultaneously, is of extreme importance in modern learning scenarios. Binary Relevance (Zhang et al., 2018) is one of the most popular approaches to MLC tasks but it fails to capture the dependency among the labels. Classifier Chains (Read et al., 2011) transforms the problem into multi-class classification task but too much computationally exhaustive. There are some loss functions (Zhang & Zhou, 2006) that have been proposed specifically for MLC tasks with DNNs but the question about including the information of label dependency remained mostly unanswered. Also these methods cannot handle data when none of the labels are present. Methods like CCA (Hardoon et al., 2004), DCCA (Andrew et al., 2013) claims to model the dependency among labels but statistical properties of these methods are still not widely understood. In this work we focus on designing a strictly proper loss which considers dependency among labels to be used during training so that weights of DNNs encodes the information of label dependency during training process.

## 3 PRELIMINARIES

**Notations:** Let $\mathcal{D} = (\mathcal{X}, \mathcal{Y})$ be a multi-label dataset sampled from a distribution $P$ on $\mathcal{X} \times \mathcal{Y}$ where $\mathbf{x} \in X$, $\mathbf{y} = [y^{(1)}, \ldots, y^{(K)}] \in \mathcal{Y}$ and $\mathcal{Y} \subset \{-1, +1\}^K$. Let $\mathcal{H}$ be the class of hypothesis that are to be considered. The task of MLC is to learn a classifier $\boldsymbol{h} \in \mathcal{H} : \mathcal{X} \to \mathbb{R}^K$ such that the classifier assigns each sample a set of labels. For MLC tasks, DNNs usually predict a set of vector values function $\boldsymbol{h} = (h_1, \ldots, h_K) : \mathcal{X} \to \mathbb{R}$. These notations are to be followed throughout the paper unless explicitly specified.

Given an observation $\mathbf{x} \in \mathcal{X}$, the goal of calibrated multi-label learning is to predict the vector of posterior probability $\rho(\mathbf{x}) = [\rho^{(1)}(\mathbf{x}), \ldots, \rho^{(K)}(\mathbf{x})]^T$ accurately

$$\rho^{(k)}(\mathbf{x}) = P(y^{(k)} = 1 | \mathbf{x}) \; ; \; \forall k \in \{1, \ldots K\}$$

To estimate the posterior probability of a label, a multi-label DNN maps the input to an embedding $\boldsymbol{h}(\mathbf{x}) = [h_1(\mathbf{x}), \ldots, h_K(\mathbf{x})]$. Then each $h_k(\mathbf{x})$ is mapped to a label probability estimate through an inverse link function (Masnadi-Shirazi & Vasconcelos, 2010) $\Psi^{-1} : \mathbb{R} \to [0, 1]$. So,

$$\hat{\rho}^{(k)}(\mathbf{x}) = P(\hat{y}^{(k)} = 1 | \mathbf{x}) = \Psi^{-1}(h_k(\mathbf{x})) \; ; \; \forall k \in \{1, \ldots, K\}$$

**Proper Loss:** Let $l$ be a multi-label loss function such that $l : [0, 1]^K \times \mathcal{Y} \to \mathbb{R}_{\geq 0}$. Given a loss $l$ the cost associated with predicting $\hat{\rho}(\mathbf{x})$ is $l(\hat{\rho(\mathbf{x})}, \pm 1)$. The goal is to produce probability estimates

that minimizes the total risk $\mathcal{R}(\rho) = E_{\mathbf{x},\mathbf{y}}[l(\rho(\mathbf{x}), \mathbf{y})]$. Multi-label DNNs try to empirically estimate $\mathcal{R}(\rho)$ during training. This can be written as

$$\mathcal{R}(\hat{\rho}; \mathcal{D}) = E_{\mathbf{x}}\left[E_{\mathbf{y}|\mathbf{x}}\left[\sum_{k=1}^{K} l(\hat{\rho}^{(k)}(\mathbf{x}), y^{(k)})|\mathbf{x}\right]\right]$$

$$= E_{\mathbf{x}}\left[\sum_{k=1}^{K} S(\hat{\rho}^{(k)}(\mathbf{x}), \rho^{(k)}(\mathbf{x}))\right] \tag{1}$$

$S(.,.)$ is the conditional risk.

**Definition 1** (Strictly Proper (Gneiting & Raftery, 2007)). *The conditional risk $S(\hat{\rho}, \rho)$ is said to be strictly proper if it is uniquely minimized by the true proper estimation i.e. $\hat{\rho} = \rho$.*

Therefore to calibrate a DNN for MLC tasks a strictly proper scoring rule is desired as a loss function so that the estimate would be the actual posterior probability.

## 4 CORRELATED MULTI-LABEL LOSS

To take into account the information of label interdependence in MLC tasks we consider correlation among pairs of labels. $\boldsymbol{h}(\mathcal{X})$ denotes the score matrix and $Y$ denotes the ground truth matrix. For both $\boldsymbol{h}(\mathcal{X})$ and $Y$ the rows and columns represent the number of labels and number of instances respectively. Our intuition is that the absolute difference between the empirical correlation($\tau$) of any two rows of $\boldsymbol{h}(\mathcal{X})$ and the same rows of $Y$, i.e. $|\tau\left(\boldsymbol{h}^{(i)}(\mathcal{X}), \boldsymbol{h}^{(j)}(\mathcal{X})\right) - \tau\left(Y^{(i)}, Y^{(j)}\right)|$ should be minimal in the ideal situation.

**Lemma 1.** *Let $\mathcal{Z} = (\mathcal{E}_i, Y_i)_{i=1}^n$ be the set of score vectors and ground truth. Also assume that predicted labels for $\mathcal{Z}$ are all correct. Consider a dataset $\mathcal{Z}'$ such that $\mathcal{Z}' = \{(\mathcal{E}_1, Y_1), \ldots, (\mathcal{E}_j, Y_j'), \ldots, (\mathcal{E}_n, Y_n)\}$ where $Y_j' = -Y_j$. Let $\mathcal{E} = [\mathcal{E}_1, \ldots, \mathcal{E}_n]$ ; $Y = [Y_1, \ldots, Y_n]$ ; $Y' = [Y_1, \ldots, Y_j', \ldots, Y_n]$ ; $\mathcal{E}^{(m)}$, $Y^{(m)}$ and $Y'^{(m)}$ are the $m^{th}$ row of $\mathcal{E}$, $Y$ and $Y$ respectively. Then*

$$\left|\tau\left(\phi \circ \mathcal{E}^{(l)}, \phi \circ \mathcal{E}^{(m)}\right) - \tau\left(Y^{(l)}, Y^{(m)}\right)\right| \leq \left|\tau\left(\phi \circ \mathcal{E}^{(l)}, \phi \circ \mathcal{E}^{(m)}\right) - \tau\left(Y'^{(l)}, Y'^{(m)}\right)\right| \tag{2}$$

$\forall l, m \in [n]; l \neq m$ *and* $\phi : \mathbb{R} \rightarrow [-1, 1]$ *is a strictly increasing function.*

Lemma 1 implies that the difference between correlation of score vectors of two labels and their ground truth counterparts increase when the score vectors are wrongfully predicted. So, a correct model will always try to minimize this distance. Incorporating this difference into the training objective allows the model to account for label-pair dependencies, thereby making the DNN aware of the underlying dependency structure. Motivated by this observation, below we propose a novel loss function that takes into account the dependency structure during the training process. We call our loss function Correlated Multi-Label Loss(CMLL). For clarity, the loss function defined below is expressed in batch form to align with the implementation:

$$\mathcal{L}_B = \frac{1}{|B|} \sum_{n=1}^{|B|} \mathcal{L}(\hat{\rho}_n, \mathbf{y}_n)$$

$$= \frac{1}{|B|} \sum_{n=1}^{|B|} \sum_{i=1}^{K} \left[ \left\{ y_n^{(i)} \log(\hat{\rho}_n^{(i)}) + (1 - y_n^{(i)})(\log(1 - \hat{\rho}_n^{(i)})) \right\} \right.$$

$$\left. + \lambda \cdot \left\{ \sum_{j=i+1}^{K} |\tau(\phi \circ \mathcal{E}_B^{(i)}, \phi \circ \mathcal{E}_B^{(j)}) - \tau(Y_B^{(i)}, Y_B^{(j)})| \right\} \right] \tag{3}$$

where $\lambda$ is a hyper-parameter and $B$ is the batch.

The first component of the loss function accounts for the presence or absence of each label in an instance but does not capture how the presence or absence of a given label influences other labels.

The second term in the right hand side of equation 3 models these interactions, thereby incorporating label dependencies into the training process and enabling the DNN to learn the underlying dependency structure. By lemma 1 we can conclude that during training process the second term in the right hand side of the equation gets minimized. Below we shall show the Strict Properness of CMLL

**Theorem 1.** *For $\lambda \in \mathbb{R}_+$ the Class Probability Estimate loss function defined in equation 3 is strictly proper and Fisher Consistent.*

Our proposed loss function is *Strictly Proper* while explicitly accounting for label interdependencies. Fisher consistency implies that a model trained with our loss function can asymptotically attain the best prediction performance. To the best of our knowledge, CMLL is the first loss function tailored for multi-label classification tasks that simultaneously satisfies both of these properties.

## 5    GENERALIZATION ANALYSIS

Understanding the generalization ability of learning algorithms- that is performance of learned machines on unseen datasets is an important question in the theoretical research in machine learning. This challenge in equally relevant in case of multi-label learning as explaining why multi-label models generalize has got only a little attention. For generalization analysis of models trained using CMLL we follow Zhang & Zhang (2024) and use Rademacher Complexity. Below we propose some definitions that we need to proceed in this section.

**Definition 2.** *(Rademacher Complexity) Let $\mathcal{H}$ be a class of real valued functions from $\mathcal{X}$ to $\mathbb{R}$. Let $D = \{\boldsymbol{x}_1, \ldots, \boldsymbol{x}_n\}$ be the set of i.i.d. random vectors. The empirical Rademacher Complexity over $\mathcal{H}$ is defined as follows:*

$$\hat{\mathfrak{R}}_D(\mathcal{H}) = E_\epsilon \left[ \sup_{h \in \mathcal{H}} \frac{1}{n} \sum_{i=1}^{n} \epsilon_i h(\boldsymbol{x}_i) \right]$$

*where $\epsilon_1, \ldots, \epsilon_n$ are Rademacher random variables.*

But in multi-label learning $\mathcal{H}$ is a class of vector valued functions. So the definition 2 of Rademacher complexity fails in case of MLC tasks. For multi-label learning problems it is a common practice to use multi-label Rademacher Complexity (Zhang & Zhang, 2024) to bound the Rademacher Complexity of a loss function space of a composed class through vector-concentration inequality Maurer (2016). The multi-label Rademacher complexity is defined as follows:

**Definition 3.** *(Multi-Label Rademacher Complexity) Let $\mathcal{H}$ be a class of vector valued functions from $\mathcal{X}$ to $\mathbb{R}^K$ and $D = \{\boldsymbol{x}_1, \ldots, \boldsymbol{x}_n\}$ be set of i.i.d. random vectors. The empirical multi-label Rademacher Complexity is defined as follows:*

$$\hat{\mathfrak{R}}_D^M(\mathcal{H}) = E_\epsilon \left[ sup_{\boldsymbol{h} \in \mathcal{H}} \frac{1}{n} \sum_{i=1}^{n} \sum_{c=1}^{K} \epsilon_{ic} h_c(\boldsymbol{x}_i) \right]$$

*where $\epsilon_{ic}$'s are independent doubly indexed Rademacher random variables.*

### 5.1    BOUNDS FOR $\ell_2$-LIPSCHITZ LOSS

For deriving generalization bounds first we make an assumptions below:

**Assumption 1.** *The loss function and the vector valued functions are bounded above: $L(.,.) \leq M$ and $|h_j(.)| \leq B$ where $B, M > 0$.*

Assumption 1 is relatively mild. Whenever we consider the hypothesis class $\mathcal{H}$ for multi-label learning, we consider that $\forall \ \boldsymbol{h} \in \mathcal{H}, \ ||\boldsymbol{h}||_2 \leq \Lambda$ (Zhang & Zhang, 2024).

**Propostion 1.** *For any $\lambda > 0$ the proposed loss fucntion CMLL defined in equation 3 is $\ell_2$-Lipschitz continuous.*

With assumption 1, proposition 1 and multi-label Rademacher Complexity we have the following theorem

**Theorem 2.** *Let $\mathcal{G}$ be the hypothesis set for MLC tasks. Let assumption 1 holds. The given dataset is D of size $N$ and loss function (CMLL) as defined in equation 3. Then for any $\delta > 0$, with probability $1 - \delta$ it can be written that:*

$$R(\boldsymbol{g}) \leq \hat{R}_D(\boldsymbol{g}) + \frac{2\sqrt{2}\mu K B}{\sqrt{N}} + 3M\sqrt{\frac{\log(2/\delta)}{2N}}$$

*$\mu$ is the Lipschitz constant and $\boldsymbol{g} = \Psi^{-1} \circ \boldsymbol{h}$*

Proof of all theorems are provided in the appendix.

In Theorem 2, we can see there is linear dependency on the number of labels $K$. Theorem 2 provides good generalization when the number of samples $(\sqrt{N})$ is larger than the number of labels because then the upper bound will move towards 0 when $N \to \infty$. In all the datasets used for our experiments, the number of samples greatly exceeds the number of labels.

## 6 EXPERIMENTS AND RESULTS

**Networks and Datasets.** To evaluate the benefits of the proposed loss function, we conducted experiments on three publicly available datasets – PASCAL VOC, MSCOCO, and WIDER-A to test the effectiveness of our models. A brief summary of the datasets is outlined below:

- PASCAL VOC 2012 (Everingham et al., 2012): The PASCAL VOC 2012 dataset extends the original PASCAL VOC challenge and comprises 5717 training images and 5823 test images annotated across 20 object categories.
- MS-COCO (Lin et al., 2014): The Microsoft Common Objects in Context (MS-COCO) dataset is a large-scale benchmark in computer vision, containing 82081 training images and 40137 test images. Unlike traditional single-label datasets, MS COCO reflects real-world scenarios by including images with complex scenes and multiple objects from 80 object categories.
- WIDER-A (Li et al., 2016): The WIDER Attribute (WIDER-A) dataset is a large-scale benchmark designed for human attribute recognition. It consists of images collected from diverse scenarios with significant variations in pose, appearance, illumination, and occlusion. The dataset contains 28345 train images and 29179 test images. Each person instance in the dataset is annotated with 14 attributes, covering aspects such as clothing style, accessories, and physical characteristics.

The proposed loss function is tested on both CNN-based models (ResNet-50 (He et al., 2016)) and transformer-based models (ViT-B/32 (Dosovitskiy et al., 2020)). For the multi-label classification task, the standard softmax prediction head of the models is replaced with a sigmoid activation function to allow independent probability estimation for each class.

**Baselines.** We use BCE, Focal Loss (FL) (Mukhoti et al., 2020), ASY (Ridnik et al., 2021) and three recently proposed losses TWL (Kobayashi, 2023), LDACE-CCL (Pal et al., 2025), SPA (Cheng & Vasconcelos, 2024) to compare our result with. Among all these losses only BCE is a proper loss. The parameters are selected as proposed in their respective original works.

**Evaluation Metrics.** To measure the accuracy in classification tasks we follow Gao & Zhou (2011) and Ridnik et al. (2021) and use Hamming Loss (HL) and mean Average Precision (mAP). To measure calibration of DNNs trained with aforementioned losses, we employ two metrics based on reliability diagram (DeGroot & Fienberg, 1983; Niculescu-Mizil & Caruana, 2005) – Average Calibration Error (ACE) (Neumann et al., 2018) and Maximum Calibration Error (MCE) (Naeini et al., 2015).

**Implementation Details.** Both the ResNet-50 and the ViT-B/32 models, together with the loss functions, are implemented in Pytorch and are trained using a single NVIDIA A6000 GPU. The ResNet-50 model is optimized using Adam optimizer with a learning rate of $1e-4$ and weight decay of $1e-5$. The ViT B/32 model is trained using AdamW optimizer, employing the same

learning rate of $1e - 4$ and a weight decay of $1e - 2$. The input image resolution for the models is set to $224 \times 224$. All the models are trained for $100$ epochs with a batch size of $32$. For evaluation, we select the model corresponding to the best validation loss. For all the experiments, the value of $\lambda$ in CMLL loss (Equation 3) is set to $1$.

**Results and Discussion.** Table 1, 2, 3 summarizes our results. We can see that our proposed loss CMLL performs better in all three cases in terms of producing posterior probabilities. The results of ACE and MCE endorses that. BCE, ASY and SPA performs worse than CMLL because none of them cannot capture the dependency between the labels. This result supports our argument that incorporating label interdependencies is essential for achieving better calibration. FL, TWL and LDACE-CCL both performs far worse in terms of ACE and MCE, owing to the fact that none of them are strictly proper. This observation endorses our initial claim that strictly proper is an essential property for loss functions in case of calibration. In terms of HL and mAP, CMLL performs well in most of the cases. This implies that CMLL does not affect the accuracy to produce well calibrated posterior probabilities.

In the tables below, the best performances are highlighted in bold. Across all three datasets and both architectures, CMLL consistently outperforms other loss functions in terms of calibration while maintaining competitive accuracy. For the PASCAL-VOC 2012 dataset, training ViT-B/32 with CMLL yields the lowest ACE and MCE values of 0.0247 and 0.0761, respectively, representing a significant improvement. The HL and mAP also achieves accuracy mostly better best result. On the MS-COCO dataset, both Resnet-50 and ViT-B/32 show substantial gains in calibration when trained with CMLL, without compromising accuracy. Similarly, on WIDER-A, ViT trained with CMLL demonstrates remarkable improvements in both calibration and accuracy.

| Model | Metric | Method | | | | | | |
|---|---|---|---|---|---|---|---|---|
| | | BCE | TWL | FL | ASY | SPA | LDACE - CCL | CMLL (Ours) |
| RN50 | H. Loss ↓ | 0.0663 | 0.1142 | 0.0687 | 0.1239 | 0.0845 | 0.0711 | **0.0219** |
| | mAP ↑ | 0.9025 | 0.8105 | **0.9273** | 0.8853 | 0.9152 | 0.9045 | 0.9251 |
| | ACE ↓ | 0.1533 | 0.3363 | 0.1403 | 0.2595 | 0.2807 | 0.2514 | **0.1265** |
| | MCE ↓ | 0.3215 | 0.4703 | 0.2825 | 0.3804 | 0.4287 | 0.4002 | **0.2189** |
| ViT | H. Loss ↓ | 0.1063 | 0.1890 | 0.1038 | 0.1520 | 0.1248 | **0.0745** | 0.0761 |
| | mAP ↑ | 0.8972 | 0.8918 | 0.9240 | 0.8983 | 0.9158 | 0.9274 | **0.9368** |
| | ACE ↓ | 0.0824 | 0.2284 | 0.1197 | 0.2917 | 0.3182 | 0.3992 | **0.0247** |
| | MCE ↓ | 0.2346 | 0.3563 | 0.2773 | 0.3411 | 0.4867 | 0.2132 | **0.0761** |

Table 1: The performance results corresponding to the PASCAL VOC 2012 dataset.

| Model | Metric | Method | | | | | | |
|---|---|---|---|---|---|---|---|---|
| | | BCE | TWL | FL | ASY | SPA | LDACE - CCL | CMLL (Ours) |
| RN50 | H. Loss ↓ | 0.0324 | 0.0479 | **0.0214** | 0.0271 | 0.0264 | 0.0284 | 0.0241 |
| | mAP ↑ | 0.9385 | 0.9184 | 0.9005 | 0.9406 | 0.9461 | 0.9666 | **0.9698** |
| | ACE ↓ | 0.0654 | 0.2057 | 0.1254 | 0.2137 | 0.1277 | 0.1385 | **0.0030** |
| | MCE ↓ | 0.1766 | 0.4170 | 0.2366 | 0.3125 | 0.1907 | 0.2347 | **0.0052** |
| ViT | H. Loss ↓ | 0.0324 | 0.1156 | 0.0295 | 0.0590 | 0.1159 | 0.0306 | **0.0310** |
| | mAP ↑ | 0.9207 | 0.9253 | 0.9147 | 0.9417 | 0.9245 | 0.9531 | **0.9775** |
| | ACE ↓ | 0.0829 | 0.2573 | 0.1143 | 0.2137 | 0.1584 | 0.1582 | **0.0020** |
| | MCE ↓ | 0.2426 | 0.4371 | 0.2607 | 0.3925 | 0.2935 | 0.2809 | **0.0038** |

Table 2: The performance results corresponding to the MS-COCO dataset.

| Model | Metric | Method | | | | | | |
|---|---|---|---|---|---|---|---|---|
| | | BCE | TWL | FL | ASY | SPA | LDACE - CCL | CMLL (Ours) |
| RN50 | H. Loss ↓ | 0.1979 | 0.1753 | 0.1912 | 0.2248 | 0.2129 | 0.2044 | **0.1741** |
| | mAP ↑ | 0.8198 | 0.6334 | 0.8087 | 0.8195 | 0.8049 | 0.7839 | **0.8248** |
| | ACE ↓ | 0.1352 | 0.2194 | 0.1235 | 0.2729 | 0.2538 | 0.2480 | **0.1232** |
| | MCE ↓ | 0.3209 | 0.3995 | 0.2492 | 0.3255 | 0.3215 | 0.3773 | **0.2256** |
| ViT | H. Loss ↓ | **0.1891** | 0.3312 | 0.2067 | 0.1976 | 0.1910 | 0.2026 | 0.1936 |
| | mAP ↑ | 0.8180 | 0.7546 | 0.7932 | 0.7935 | 0.7921 | 0.7919 | **0.8226** |
| | ACE ↓ | 0.0795 | 0.2787 | 0.2124 | 0.2301 | 0.2214 | 0.1864 | **0.0145** |
| | MCE ↓ | 0.2783 | 0.3768 | 0.2824 | 0.2885 | 0.3215 | 0.3051 | **0.0233** |

Table 3: The performance results corresponding to the WIDER-A dataset.

## 7 CONCLUSION

In this paper we introduced CMLL, a novel loss function to calibrate MLC tasks while maintaining competitive accuracy. CMLL is strictly proper and explicitly accounts for label inter dependencies, providing reliable posterior probability estimates. We provided a theoretical justification for its formulation, proved its Fisher consistency, and derived a generalization bound showing linear dependency on the number of labels. In future work, we plan try to explore the performance of CMLL in under co-variate shifts, out of the domain settings as well as tighter generalization bound of our loss function.

## DECLARATION

LLMs were used solely to polish the English language in this paper. All technical content, ideas, results, and analyses are the original work of the authors.

## REPRODUCIBILITY STATEMENT

We are committed to ensuring the reproducibility of our results. To this end, we provide the following:

- Code and Implementation Details: All models, loss functions, and training pipelines are implemented in PyTorch (Ref. Section 6). The source code, including data preprocessing scripts, model definitions, and training codes are given in `https://github.com/medprocguy/CMLL`.
- Datasets: We use only publicly available datasets — PASCAL VOC, MS-COCO, and WIDER Attribute — and have used the training and test splits provided in the dataset.
- Hyperparameters: Detailed hyperparameter settings (e.g., learning rates, weight decay, optimizers, batch sizes, and number of epochs) are reported in Section 6.
- Experimental Setup: All experiments are conducted on a single NVIDIA A6000 GPU. The selection criterion for final models is based on the best validation loss.
- Randomness and Seeds: We fix random seeds for model initialization and training to ensure consistent results across runs.

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

APPENDIX

**Proof of Lemma 1**

*Proof.* Assume that the predicted labels form the score vectors are all correct for $D = (\mathcal{E}_i, Y_i)_{i=1}^n$. Define $D'$ as stated.
For ease of notation we define $e^{(l)} = \phi \circ \mathcal{E}^{(l)}$ and $\overline{e^{(l)}} = mean(\phi \circ \mathcal{E}^{(l)})$ same goes for index $m$.

$$
\left| \tau\left(\phi \circ \mathcal{E}^{(l)}, \phi \circ \mathcal{E}^{(m)}\right) - \tau\left(Y^{(l)}, Y^{(m)}\right) \right| - \left| \tau\left(\phi \circ \mathcal{E}^{(l)}, \phi \circ \mathcal{E}^{(m)}\right) - \tau\left(Y'^{(l)}, Y'^{(m)}\right) \right|
$$

$$
= \left| \left( \frac{(e^{(l)} - \overline{e^{(l)}})(e^{(m)} - \overline{e^{(m)}})^T}{\sqrt{\mathrm{Var}(e^{(l)})} \cdot \sqrt{\mathrm{Var}(e^{(m)})}} - \frac{(Y^{(l)} - \overline{Y^{(l)}})(Y^{(m)} - \overline{Y^{(m)}})^T}{\sqrt{\mathrm{Var}(Y^{(l)})} \cdot \sqrt{\mathrm{Var}(Y^{(m)})}} \right) \right| -
$$

$$
\left| \left( \frac{(e'^{(l)} - \overline{e'^{(l)}})(e'^{(m)} - \overline{e'^{(m)}})^T}{\sqrt{\mathrm{Var}(e'^{(l)})} \cdot \sqrt{\mathrm{Var}(e'^{(m)})}} - \frac{(Y'^{(l)} - \overline{Y'^{(l)}})(Y'^{(m)} - \overline{Y'^{(m)}})^T}{\sqrt{\mathrm{Var}(Y'^{(l)})} \cdot \sqrt{\mathrm{Var}(Y'^{(m)})}} \right) \right|
$$

$$(4)$$

choose,

$$
M = \min\{ \sqrt{\mathrm{Var}(e^{(l)})}, \sqrt{\mathrm{Var}(e^{(m)})}, \sqrt{\mathrm{Var}(e'^{(l)})}, \sqrt{\mathrm{Var}(e'^{(m)})},
$$

$$
\sqrt{\mathrm{Var}(Y^{(l)})}, \sqrt{\mathrm{Var}(Y^{(m)})}, \sqrt{\mathrm{Var}(Y'^{(l)})}, \sqrt{\mathrm{Var}(Y'^{(m)})} \}
$$

Now by replacing $M$ in the expression above we get that

$$
\left| \tau\left(\phi \circ \mathcal{E}^{(l)}, \phi \circ \mathcal{E}^{(m)}\right) - \tau\left(Y^{(l)}, Y^{(m)}\right) \right| - \left| \tau\left(\phi \circ \mathcal{E}^{(l)}, \phi \circ \mathcal{E}^{(m)}\right) - \tau\left(Y'^{(l)}, Y'^{(m)}\right) \right|
$$

$$
\leq \frac{1}{M}\Big[ \left| \left( (e^{(l)} - \overline{e^{(l)}})(e^{(m)} - \overline{e^{(m)}})^T - (Y^{(l)} - \overline{Y^{(l)}})(Y^{(m)} - \overline{Y^{(m)}})^T \right) \right|
$$

$$
- \left| \left( (e'^{(l)} - \overline{e'^{(l)}})(e'^{(m)} - \overline{e'^{(m)}})^T - (Y'^{(l)} - \overline{Y'^{(l)}})(Y'^{(m)} - \overline{Y'^{(m)}})^T \right) \right| \Big]
$$

The expression above tells us about how close the inner products are. So we can easily conclude that for the dataset $D$ the 1st term inside the expression is closer to the inner product of the ground truths as for $D$ all the predicted labels are correct. Hence, lemma 1 is proved. $\square$

**Proof of Theorem 1**

*Proof.* From equation 1 we have,

$$
\mathcal{R}(\hat{\rho}; \mathcal{D}) = E_{\mathbf{x}}\left[ \sum_{k=1}^{K} S(\hat{\rho}^{(k)}(\mathbf{x}), \rho^{(k)}(\mathbf{x})) \right] \tag{5}
$$

Equation 3 represents the empirical loss. So, by replacing it with the right hand side of equation 5 we get that

$$
\mathcal{R}(\hat{\rho}; \mathcal{D}) = \frac{1}{N} \sum_{n=1}^{N} \sum_{i=1}^{K} \Bigg[ \left\{ \rho_n^{(i)} \log(\hat{\rho}_n^{(i)}) + (1 - \rho_n^{(i)})(\log(1 - \hat{\rho}_n^{(i)})) \right\}
$$

$$
+ \lambda \cdot \left\{ \sum_{j=i+1}^{K} |\tau(\hat{\rho}^{(i)}, \hat{\rho}^{(i)}) - \tau(\rho^{(i)}, \rho^{(i)})| \right\} \Bigg] \tag{6}
$$

So, we can see that $\mathcal{R}(\hat{\rho}; \mathcal{D})$ can only be minimized if and only if $\hat{\rho} = \rho$. As, $\mathcal{R}(\hat{\rho}; \mathcal{D}) \geq 0$ and $S(\hat{\rho}^{(k)}(\mathbf{x}), \rho^{(k)}(\mathbf{x})) \geq 0$ it can be said that $S(\rho, \hat{\rho})$ is minimized uniquely when $\hat{\rho} = \rho$.

Fisher Consistency of CMLL is a direct consequence of being a Strictly proper loss (Reid & Williamson, 2010). □

**Proof of Theorem 2:** We follow the style of proof by Zhang & Zhang (2024) and Maurer (2016) to prove our theorem.

*Proof.* Define, $\boldsymbol{g} = \Psi^{-1} \circ \boldsymbol{h}$ ; $\mathcal{L} = \{L(\boldsymbol{g}(\mathbf{x}), \mathbf{y}) | \boldsymbol{g} \in \}$ where $L$ is CMLL. $L$ can be written as $L = C + \lambda \cdot \Theta$.

Let $D = \{(\mathbf{x}_i, \mathbf{y}_i) \mid i \in [N]\}$. Define $D'$ such that only one sample is different than $D$. Assume that $m^{th}$ sample in $D$ is replaced with $(\mathbf{x}'_m, \mathbf{y}'_{\mathbf{m}})$.

Define, $\Phi(D) = \sup_{g \in \mathcal{G}} [E_{\mathbf{x}, \mathbf{y}}[L(\boldsymbol{g}(\mathbf{x}), \mathbf{y})] - \frac{1}{N} \sum_{i=1}^{N} L(\boldsymbol{g}(\mathbf{x}_i), \mathbf{y}_i)] = \sup_{g \in \mathcal{G}} [C(\boldsymbol{g}) + \lambda \cdot \Theta(\boldsymbol{g}) - \hat{C}_D(\boldsymbol{g}) - \lambda \cdot \hat{\Theta}_D(\boldsymbol{g})]$.

$\Phi(D') - \Phi(D)$

$= \sup_{\boldsymbol{g} \in \mathcal{G}} [C(\boldsymbol{g}) + \lambda \cdot \Theta(\boldsymbol{g}) - \hat{C}_{D'}(\boldsymbol{g}) - \lambda \cdot \hat{\Theta}_{D'}(\boldsymbol{g})] - \sup_{g \in \mathcal{G}} [C(\boldsymbol{g}) + \lambda \cdot \Theta(\boldsymbol{g}) - \hat{C}_D(\boldsymbol{g}) - \lambda \cdot \hat{\Theta}_D(\boldsymbol{g})]$

$\leq \sup_{\boldsymbol{g} \in \mathcal{G}} [\hat{C}_D(\boldsymbol{g}) + \lambda \cdot \hat{\Theta}_D(\boldsymbol{g}) - \hat{C}_{D'}(\boldsymbol{g}) - \lambda \cdot \hat{\Theta}_{D'}(\boldsymbol{g})]$

$= \sup_{\boldsymbol{g} \in \mathcal{G}} \left( \frac{L(\boldsymbol{g}(\mathbf{x}_m), \mathbf{y}_m) - L(\boldsymbol{g}(\mathbf{x}'_m), \mathbf{y}'_m)}{N} \right)$

$\leq \frac{M}{N}$

From McDiarmid's inequality (Zhang & Zhang, 2024), for any $0 < \delta < 1$ with probability at least $1 - \delta/2$ it can be written for any dataset $D$ that

$$\Phi(D) \leq \mathbb{E}_D[\Phi(D)] + M \sqrt{\frac{\ln(2/\delta)}{2N}}. \tag{7}$$

We need to estimate the upper bound of $\mathbb{E}_D[\Phi(D)]$

$\mathbb{E}_D[\Phi(D)]$

$= E_D[\sup_{\boldsymbol{g} \in \mathcal{G}} [E_{D'}[\hat{C}_{D'}(\boldsymbol{g}) + \lambda \cdot \hat{\Theta}_{D'}(\boldsymbol{g}) - \hat{C}_D(\boldsymbol{g}) + \lambda \cdot \hat{\Theta}_D(\boldsymbol{g})]]]$

$= E_{D, D'}[\sup_{\boldsymbol{g} \in \mathcal{G}} [E_{D'}[\hat{C}_{D'}(\boldsymbol{g}) + \lambda \cdot \hat{\Theta}_{D'}(\boldsymbol{g}) - \hat{C}_D(\boldsymbol{g}) + \lambda \cdot \hat{\Theta}_D(\boldsymbol{g})]]$

$= E_{D, D'}[\sup_{\boldsymbol{g} \in \mathcal{G}} \frac{1}{N} \sum_{i=1}^{N} [L(g(\mathbf{x}'_i), \mathbf{y}'_i) - L(g(\mathbf{x}_i), \mathbf{y}_i)]]$

$= E_{\boldsymbol{\epsilon}, D, D'}[\sup_{\boldsymbol{g} \in \mathcal{G}} \frac{1}{N} \sum_{i=1}^{N} \epsilon_i [L(g(\mathbf{x}'_i), \mathbf{y}'_i) - L(g(\mathbf{x}_i), \mathbf{y}_i)]]$

because $\mathbb{P}(\epsilon_i = +1) = \mathbb{P}(\epsilon_i = -1) = \frac{1}{2}$ is a Rademacher random variable

$\leq E_{\boldsymbol{\epsilon}, D'}[\sup_{\boldsymbol{g} \in \mathcal{G}} \frac{1}{N} \sum_{i=1}^{N} \epsilon_i [L(g(\mathbf{x}'_i), \mathbf{y}'_i)]] + E_{\boldsymbol{\epsilon}, D}[\sup_{\boldsymbol{g} \in \mathcal{G}} \frac{1}{N} \sum_{i=1}^{N} -\epsilon_i [L(g(\mathbf{x}_i), \mathbf{y}_i)]]$

$= 2 E_{\boldsymbol{\epsilon}, D}[\sup_{\boldsymbol{g} \in \mathcal{G}} \frac{1}{N} \sum_{i=1}^{N} \epsilon_i [L(g(\mathbf{x}_i), \mathbf{y}_i)]] \tag{8}$

From McDiarmid's inequality we get

$$E_{\boldsymbol{\epsilon},D}[\sup_{\boldsymbol{g}\in\mathcal{G}}\frac{1}{N}\sum_{i=1}^{N}\epsilon_i[L(g(\mathbf{x}_i),\mathbf{y}_i)]] \leq E_{\boldsymbol{\epsilon}}[\sup_{\boldsymbol{g}\in\mathcal{G}}\frac{1}{N}\sum_{i=1}^{N}\epsilon_i[L(g(\mathbf{x}_i),\mathbf{y}_i)]] + M\sqrt{\frac{\ln(2/\delta)}{2N}}$$

So,

$$\mathfrak{R}(\mathcal{L}) \leq \hat{\mathfrak{R}}_D(\mathcal{L}) + M\sqrt{\frac{\ln(2/\delta)}{2N}} \tag{9}$$

By combining equations 7, 8 and 9 we get that

$$R(\boldsymbol{g}) \leq \hat{R}_D(\boldsymbol{g}) + 2\hat{\mathfrak{R}}_D(\mathcal{L}) + 3M\sqrt{\frac{\ln(2/\delta)}{2N}}. \tag{10}$$

As our loss function is $\mu$-Lipschitz, from corollary 1 in Maurer (2016) we have

$$E_{\boldsymbol{\epsilon}}[\sup_{\boldsymbol{g}\in\mathcal{G}}\frac{1}{N}\sum_{i=1}^{N}\epsilon_i L(\boldsymbol{g}(\mathbf{x}_i),\mathbf{y}_i)] \leq \sqrt{2}\mu E_{\boldsymbol{\epsilon}}[\sup_{\boldsymbol{g}\in\mathcal{G}}\frac{1}{N}\sum_{i=1}^{N}\sum_{l=1}^{K}\epsilon_{il}g_l(x_i)] \tag{11}$$

By using equation 11 we have:

$$\hat{\mathfrak{R}}_D(\mathcal{L}) \leq \sqrt{2}\mu E_{\boldsymbol{\epsilon}}[\sup_{\boldsymbol{g}\in\mathcal{G}}\frac{1}{N}\sum_{i=1}^{N}\sum_{l=1}^{K}\epsilon_{il}g_l(x_i)]$$

$$\leq \sqrt{2}\mu K \max_l E_{\boldsymbol{\epsilon}}[\sup_{\boldsymbol{g}_l\in\mathcal{G}_l}\frac{1}{N}\sum_{i=1}^{N}\epsilon_{il}g_l(x_i)]$$

By using Khintchine-Kahane inequality (Lust-Piquard & Pisier, 1991) with $p > 1$ we have

$$\leq \sqrt{2}\mu K \max_l \sup_{\boldsymbol{g}_l\in\mathcal{G}_l}\frac{1}{N}(\sum_{i=1}^{N}(g_l(x_i))^2)^{\frac{1}{2}}$$

By using assumption 1 we have

$$\leq \frac{\sqrt{2}\mu K \max_l B}{\sqrt{N}}$$

Combining this with equation 10 we get that

$$R(\boldsymbol{g}) \leq \hat{R}_D(\boldsymbol{g}) + \frac{2\sqrt{2}\mu KB}{\sqrt{N}} + 3M\sqrt{\frac{\log(2/\delta)}{2N}}$$

.

$\square$

