# OpenReview forum: "Leveraging Label Dependencies for Calibration in Multi-Label Classification through Proper Scoring Rule"
_ICLR.cc/2026/Conference — Submitted to ICLR 2026_

### Official Review · Reviewer_8TpF · 2025-10-26

**Soundness:** 2
**Presentation:** 1
**Contribution:** 2
**Rating:** 2
**Confidence:** 3

**Summary:**

This paper proposes Correlated Multi-Label Loss (CMLL), a novel loss function designed to improve the calibration of deep neural networks in multi-label classification (MLC) tasks. Unlike conventional losses such as Binary Cross-Entropy, which assume label independence, CMLL explicitly models pairwise label dependencies while maintaining the property of being strictly proper, ensuring reliable posterior probability estimates. The authors provide theoretical guarantees, proving that CMLL is both Fisher consistent and $\ell_2$-Lipschitz continuous, and they derive a generalization bound that scales linearly with the number of labels. Extensive experiments on benchmark datasets (PASCAL VOC 2012, MS-COCO, and WIDER-A) demonstrate that CMLL significantly reduces calibration errors without compromising classification accuracy, establishing it as an effective and theoretically grounded approach for trustworthy multi-label learning.

**Strengths:**

1. The proposed Correlated Multi-Label Loss (CMLL) innovatively combines pairwise label dependency modeling with the property of strict properness, bridging a clear gap between calibration theory and multi-label learning.

2. The paper provides formal proofs showing that CMLL is strictly proper, Fisher consistent, and $\ell_2$-Lipschitz continuous, and it derives a generalization bound with interpretable dependence on the number of labels.

**Weaknesses:**

1. The notation in this paper could be made clearer. For example, when describing $\boldsymbol{h}(\mathcal{X})$ and $Y$, it would be helpful to explicitly clarify what their rows and columns represent;

2. Although Lemma 1 seems intended to express the difference between two labels, based on my understanding of the notation, the computation of $\tau$ appears to measure the discrepancy between two instances rather than between a pair of labels.

3. In Assumption 1, it seems that the loss function $L$ corresponds to the proposed CMLL loss. If this is the case, it would be helpful to specify the valid ranges of $M$ and $B$. In particular, under certain extreme cases, the $\log$ term in Equation (3) might lead to an unbounded $M$, which could invalidate Assumption 1 and consequently affect the soundness of Theorem 2.

**Questions:**

Please carefully check the Weaknesses.

Minor comment:

1. In Lemma 1, could the authors clarify whether the dataset $\mathcal{D}$ is defined as D = \{(\varepsilon_i, Y_i)\}_{i=1}^n?

---

> ### Author Response · Authors · 2025-11-19
>
> We thank the reviewer for their valuable reviewes. The responses are provided below.
>
> $ \textbf{Q.1. The notation in this paper .... columns represent}$
>
> Ans: To address clarification regarding the rows and columns of $h(\mathcal{X})$ and $Y$, we would like to refer line 174 of the manuscript where we explicitly state: "For both $h(\mathcal{X})$ and $Y$ the rows and columns represent the number of labels and number of instances, respectively." We hope this clarifies the intended meaning and resolves the ambiguity regarding the structure of $h(\mathcal{X})$ and $Y$.
>
> $\textbf{Q.2. Although Lemma 1......a pair of labels}$
>
> Ans: To clarify, $\mathcal{E}_i$ denotes the $i^{{th}}$ column of the score matrix $\boldsymbol{h}(\mathcal{X})$, and $Y_i$ denotes the $i^{{th}}$ column of the ground-truth matrix $Y$. As stated earlier in the manuscript (line 174), the rows correspond to labels and the columns correspond to instances for both $\boldsymbol{h}(\mathcal{X})$ and $Y$.
>
> In line 180, we define $\mathcal{E}$ and $Y$ following this convention, and in line 181 we introduce $\mathcal{E}^{(m)}$ and $Y^{(m)}$ to explicitly refer to the $m^{{th}}$ row, i.e., the scores for the $m^{{th}}$ label across all instances. Consequently, $\tau\left(\phi \circ \mathcal{E}^{(l)},\, \phi \circ \mathcal{E}^{(m)}\right)$ indeed measures the discrepancy between the two labels $l$ and $m$, rather than between instances. The same interpretation applies analogously to $Y$ and $Y'$. We will refine the notations in the revised version within the rebuttal period to avoid any ambiguity.
>
> $\textbf{Q.3. In Assumption 1.....soundness of Theorem 2.}$
>
> Ans: Thank you for pointing out this concern. We clarify that in line 258, we explicitly assume that the representation scores $\boldsymbol{h}$ are bounded. This assumption is standard and can be enforced in practice using several well-known techniques, such as bounded activation functions or weight normalization. In our experiments, we employ the bounded activation function $\tanh$, which guarantees that $||\boldsymbol{h} || < \Lambda$ for some finite $\Lambda$.
>
> Consequently, each score satisfies $|h_j| \leq B$ for some finite constant $B>0$. (We acknowledge a small typo in the current manuscript where the condition was written as $h_j < B$; we will correct this to $|h_j| \leq B$ in the revised version.)
>
> Given this boundedness, the predicted probability for label $j$ satisfies
> $$
> \rho^{(j)} = \frac{1}{1 + e^{-h_j}} \leq \frac{1}{1 + e^{-B}},
> $$
> which implies
> $$
> \log \rho^{(j)} \leq \log\left( \frac{1}{1 + e^{-B}} \right).
> $$
>
> Similarly, since
> $$
> \rho^{(j)} \geq \frac{1}{1 + e^{B}},
> $$
> we obtain
> $$
> \log(1 - \rho^{(j)}) \leq \log\left( 1 - \frac{1}{1 + e^{B}} \right).
> $$
>
> Thus, both $\log \rho^{(j)}$ and $\log(1 - \rho^{(j)})$ remain finite under all extreme cases admissible by the model. As a result, the constant $M$ in Assumption 1 is guaranteed to exist and remain finite.
>
> To address the question regarding the ranges of $M$ and $B$, we would like to clarify that our analysis only requires the existence of finite constants $M$ and $B$ in order to establish the validity of Theorem 2. The precise numerical ranges of these constants are not needed; it is sufficient that they exist and are finite under the boundedness assumptions stated in the manuscript.
>
> $\textbf{Q.4. In Lemma 1, could the authors clarify whether the dataset $\mathcal{D}$ is defined as $D = (\varepsilon_i, Y_i)_{i=1}^n$?}$
>
> Ans: $\mathcal{D}$ denotes the dataset, where $\mathcal{D} = (\mathcal{X}, \mathcal{Y})$; here, $\mathcal{X}$ corresponds to the instances and $\mathcal{Y}$ corresponds to the labels. We apologize for the confusion caused by the earlier notation. In the revised version, we will replace $D$ with $\mathcal{Z}$ to improve clarity enhance the overall readability of our paper.
>
>      We sincerely thank the reviewer for their thoughtful feedback and the opportunity to address their concerns. We have responded comprehensively to each point and hope that the clarifications provided adequately resolve the issues raised. In light of these responses, we kindly request the reviewer to reconsider their evaluation and, if appropriate, provide a more favourable rating for our paper.

---

### Official Review · Reviewer_w8kB · 2025-10-31

**Soundness:** 2
**Presentation:** 2
**Contribution:** 2
**Rating:** 2
**Confidence:** 5

**Summary:**

This paper introduces a new loss function, Correlated Multi-Label Loss (CMLL), for improving confidence calibration in multi-label classification. The authors argue that existing methods either assume label independence or lack theoretical guarantees such as strict propriety. CMLL is designed to explicitly model pairwise label correlations while maintaining the property of being a Strictly Proper Scoring Rule (PSR). The paper also provides theoretical justification (Fisher consistency and generalization analysis) and experimental validation on three standard datasets (PASCAL VOC, MS-COCO, and WIDER-A), showing improvements in calibration metrics while maintaining comparable accuracy.

**Strengths:**

1. The work tackles the important and underexplored problem of multi-label confidence calibration, which is highly relevant for real-world applications where label dependencies are common (e.g., medical imaging, scene recognition).

2. Experiments across multiple datasets and architectures (ResNet-50 and ViT-B/32) demonstrate consistent improvement in calibration metrics such as ACE and MCE.

**Weaknesses:**

1. The proposed method’s originality is somewhat incremental compared to recent works such as [Chen et al., TIP 2024][Peng et al., CVPR 2024; TPAMI 2025]. These papers also explore correlation-based or dependency-aware regularization for calibration. The current submission does not clearly articulate how CMLL is fundamentally different or superior in modeling dependencies beyond reformulating correlation alignment as a proper scoring rule.

2. The experiments do not include a comparison with [Chen et al., TIP 2024], which introduced both a multi-label calibration method and comprehensive evaluation metrics for multi-label confidence calibration.

3. Only simple baselines (BCE, Focal Loss, TWL, LDACE-CCL) are used. Missing comparisons with state-of-the-art multi-label calibration methods significantly weakens the empirical validation.

4. The paper evaluates only on relatively standard architectures (ResNet-50 and ViT-B/32) and basic multi-label baselines. Recent multi-label recognition backbones (e.g., ASL [Ridnik et al., ICCV 2021], ML-Decoder, or transformer-based decoders) are not included, making it difficult to assess general applicability.

[Chen et al., TIP 2024] Dynamic Correlation Learning and Regularization for Multi-Label Confidence Calibration.

[Peng et al., CVPR 2024; TPAMI 2025] Perception/Semantic Aware Regularization for Sequential Confidence Calibration.

**Questions:**

See Weakness.

---

> ### Author Response · Authors · 2025-12-03
>
> We thank the reviewer for their comments. Our rebuttal is provided below.
>
> $\textbf{Q: The proposed method’s originality ...... as a proper scoring rule. }$
>
> $\textbf{Ans:}$ We thank the reviewer for the insightful comment. Our approach differs fundamentally from Chen et al. (TIP 2024) and Peng et al. (CVPR 2024). Specifically, our method explicitly computes the discrepancy between the correlation of the predicted score vectors for each pair of labels and their corresponding ground‐truth label correlations, and incorporates this discrepancy directly into the training objective, thereby enforcing correlation consistency and aligning the learned dependencies with the true dependency structure.
>
> In contrast, [Chen et al., TIP 2024] adopt a two-stage strategy: they first train the DCLR algorithm to produce soft labels that implicitly encode inter-label relationships based on feature-level similarities and prototype structures, and then train a multi-label recognition model using these softened targets. Their framework therefore adjusts the target labels, whereas our method directly constrains the model’s predictive behaviour through correlation matching.
>
> In [Peng et. al, CVPR 2024] the authors constructed a similarity sequence set that comprises sequences either perception similar to the sequence instance inside the input sequence or semantic correlated with the corresponding target sequence. Our method on the other hand calculates the similarity from the score vectors and ground truth itself - construction of no other similarity sequence is necessary. We hope that this clarifies the fundamental difference between our method and the two methods indicated by the reviewer.
>
> We emphasize that the proposed CMLL loss is a $\textbf{strictly proper scoring rule}$ (Theorem 1), and is therefore uniquely minimized by the true posterior probabilities, ensuring statistically valid probability estimates. It is well known that reliable probabilistic predictions can only be guaranteed when training with strictly proper losses. To the best of our knowledge, CMLL is the first loss to jointly model label dependencies while retaining strict properness, providing both principled calibration and dependency modeling. In contrast, the losses used in Chen et al. and Peng at al. has not been established as strictly proper, and its statistical guarantees for probability estimation remain unclear.
>
> We once again thank the reviewer for pointing our these two recent works related to the theme of our paper. We shall include these two works in the related works section in the revised version of our paper within the rebuttal period.
>
> $Q: \textbf{The experiments do not include ....... multi-label confidence calibration.}$
>
> $\textbf{Ans:}$ We thank the reviewer for pointing out the relevance of the recent work by Chen et al. (TIP 2024). While their method is an important contribution, it follows a fundamentally different paradigm from ours. Specifically, Chen et al. rely on a two-stage pipeline that requires CNN- and SARL-based models to first generate soft labels, which are then used to train a separate multi-label regression model.
>
> In contrast, our proposed method is fully model-agnostic and can be seamlessly integrated into any existing multi-label classification architecture using a single-stage training procedure, without requiring any auxiliary models or soft-label generation. This makes our approach considerably simpler are more general. Due to these fundamental differences in assumptions and training pipelines, a direct empirical comparison would not be strictly fair. In addition to the methods included in the earlier version of the manuscript, we have added comparisons with two [ Ridnik et al., 2021; Cheng & Vasconcelos, 2024] recent state-of-the-art methods for multi-label calibration in the revised version. Our approach achieves the best performance in terms of both classification accuracy and calibration.
>
> $\textbf{Q: The paper evaluates ...... assess general applicability.}$
>
> $\textbf{Ans:}$ Our goal in this work is to isolate and analyze the contribution of the proposed method itself, independent of architectural advances. For this reason, we intentionally selected widely adopted and well-understood backbones like ResNet-50 and ViT-B/32 to ensure fair comparability with a large body of prior multi-label calibration work that also uses these standard architectures and architectural neutrality, so that improvements can be attributed purely to our approach rather than complementary gains from more sophisticated model designs.
>
>     We sincerely thank the reviewer for their thoughtful feedback and the opportunity to address their concerns. We have responded comprehensively to each point and hope that the clarifications provided adequately resolve the issues raised. In light of these responses, we kindly request the reviewer to reconsider their evaluation and, if appropriate, provide a more favourable rating for our paper.

---

### Official Review · Reviewer_ffuK · 2025-11-01

**Soundness:** 3
**Presentation:** 3
**Contribution:** 2
**Rating:** 4
**Confidence:** 4

**Summary:**

This paper tackles the problem of poor confidence calibration in modern deep neural networks for multi-label classification tasks. This is a crucial issue, as miscalibrated models are unreliable in safety-critical applications and often involve multiple labels per instance. The authors identify a key gap in current methods: existing "proper scoring rules" (PSR) losses like BCE ignore label dependencies, and other losses that model the dependencies like focal loss are not PSR. To fix this, the paper introduces a new loss function called Correlated Multi-Label Loss (CMLL), including a regularization term penalizes the difference between the model's predicted label correlations and the ground-truth label correlations. The authors provide a key theoretical proof that their combined CMLL loss is still a PSR loss.

**Strengths:**

1. The paper is very well-motivated. Calibration is a known, hard problem in MLC. This work is well-positioned. It directly addresses the limitations of recent key papers.

2. The main claim isn't just based on intuition.

3. The experimental results are good.

**Weaknesses:**

The paper's entire theoretical foundation rests on the claim that CMLL is a PSR. A PSR must be uniquely minimized when the prediction $\hat{\rho}$ equals the true probability $\rho$. However, CMLL is a weighted trade-off between the BCE loss (a PSR) and a new correlation term. It is possible that a model will sacrifice perfect calibration (increasing the BCE loss) to better match the in-batch label correlations (decreasing the new term). This means the minimum of the CMLL loss is no longer guaranteed to be at the point of perfect calibration in particle. Since the proposed correlation term is calculated in-batch, the loss for any single sample dependent on the other samples present in its batch. This formulation contradicts the standard definition of a PSR, which is based on the expectation $E_{x,y}[L(h(x), y)]$. Also it makes the training gradient highly sensitive to batch composition and sampling noise.

The $\lambda$ is the most important part of the proposed method, as it controls the trade-off between calibration and the regularization term. The paper simply states $\lambda=1$ is used for all experiments with no justification, ablation study, or sensitivity analysis.

**Questions:**

Please refer to above

---

> ### Author Response · Authors · 2025-11-27
>
> We appreciate the reviewer’s insightful observation. We would like to clarify that Theorem 1 in our manuscript establishes that CMLL is a strictly proper scoring rule. As shown in the proof (Appendix), Equation (6) demonstrates that the correlation term in the proposed CMLL loss is also minimized by the true posterior probabilities. Consequently, the model does not trade off perfect calibration in order to fit the label correlations; both objectives are simultaneously achieved at the true posterior.
>
> We chose $\lambda = 1$ after conducting experiments with a range of values and observed that $\lambda = 1$ consistently yielded the best performance. Our intuition is that this setting allows the two components of the loss to contribute according to their inherent scales, without introducing any artificial bias toward either of the components of the CMLL loss. By choosing $\lambda = 1$, the model propagates the original ratio between these terms exactly as designed. We shall clarify this in the revised version of our paper within the rebuttal period.
>
>      We sincerely thank the reviewer for their thoughtful feedback and for giving us the opportunity to address their concerns. We have provided detailed responses to each point and hope that our clarifications satisfactorily resolve the issues raised. In light of these explanations, we respectfully request the reviewer to reconsider their evaluation and, if deemed appropriate, provide a more favourable rating for our paper.

---

### Official Review · Reviewer_gnQh · 2025-11-03

**Soundness:** 3
**Presentation:** 3
**Contribution:** 2
**Rating:** 4
**Confidence:** 5

**Summary:**

The calibration of multi-label deep neural networks is considered. The paper introduces the Correlated Multi-Label Loss (CMLL), a novel loss function designed to improve calibration in Multi-Label Classification (MLC) tasks by explicitly capturing label interdependencies. CMLL is proven to be a strictly proper loss and to be Fisher consistent. The loss incorporates dependency information by minimizing the absolute difference between the empirical correlation of the predicted scores for label pairs and the correlation of their ground truths. Extensive experiments on three benchmark datasets, PASCAL VOC, MS-COCO, and WIDER-A, demonstrate that CMLL reduces calibration error while maintaining classification accuracy compared to some other popular loss functions.

**Strengths:**

1. The work proposed the Correlated Multi-Label Loss (CMLL) and established a generalization bound for it.
2. Empirical evaluation of CMLL in terms of the accuracy and calibration on multiple real-world multi-label datasets.

**Weaknesses:**

More experiments are necessary.
   -- a. Only one metric (hamming loss) is used for evaluating the accuracy of multi-label classification. In the modern multi-label learning literature, more metrics such as mAP, OF1, CF1 are widely employed.
   -- b. Lack of comparison against state-of-the-art baselines. Comparison to SOTA multi-label losses, such as Ridnik et al., 2021; and Cheng & Vasconcelos (2024), is necessary for validating the superiority of the proposed CMLL loss against losses that do not take into consideration label dependency.

**Questions:**

There should be a space between text and Parenthesis (.

---

> ### Author Response · Authors · 2025-12-02
>
> We thank the reviewer for their comments. Our rebuttal is provided below.
>
> $Q: \textbf{Only one metric ...... are widely employed.}$
>
> $\textbf{Ans:}$ We thank the reviewer for highlighting the need for additional evaluation metrics beyond Hamming Loss. In response, we have included mAP, a widely used and interpretable metric in recent multi-label literature [Ridnik et al., 2021; Cheng & Vasconcelos, 2024]. The updated results (see table below) show that CMLL performs on par with recent state-of-the-art methods in terms of classification accuracy.
>
> | Method (mAP↑ scores)                                  |ViT  | RN50  |
> |------------------------------------------|-------|------|
> |BCE | 0.9207 | 0.9385 |
> |TWL| 0.9253| 0.9184|
> |FL| 0.9147| 0.9005|
> | ASY (Ridnik et al., 2021)                | 0.9417  | 0.9406 |
> | SPA (Cheng & Vasconcelos, 2024)          | 0.9245  | 0.9461 |
> |LDACE-CCL| 0.9531| 0.9666|
> | **CMLL (ours)**                           | **0.9775** | **0.9698** |
>
> $\textbf{mAP score corresponding to the MS-COCO dataset.}$
>
>
> | Method (mAP↑ scores)                                    |ViT  | RN50  |
> |------------------------------------------|-------|------|
> |BCE | 0.8972 | 0.9025 |
> |TWL| 0.8918 |0.8105 |
> |FL| 0.924| 0.9273|
> | ASY (Ridnik et al., 2021)                | 0.8983  | 0.8853 |
> | SPA (Cheng & Vasconcelos, 2024)          | 0.9158  | 0.9152 |
> |LDACE-CCL| 0.9274| 0.9045|
> | **CMLL (ours)**                           | **0.9368** | **0.9251** |
>
> $\textbf{mAP score corresponding to the VOC 2012 dataset.}$
>
>
> | Method (mAP↑ scores)                                    |ViT  | RN50  |
> |------------------------------------------|-------|------|
> |BCE | 0.818 | 0.8198 |
> |TWL| 0.7546 |0.6334 |
> |FL| 0.7932| 0.8087|
> | ASY (Ridnik et al., 2021)                | 0.7935  | 0.8195 |
> | SPA (Cheng & Vasconcelos, 2024)          | 0.7921  | 0.8049 |
> |LDACE-CCL| 0.7919| 0.7839|
> | **CMLL (ours)**                           | **0.8226** | **0.8248** |
>
> $\textbf{mAP score corresponding to the WIDER-A dataset.}$
>
> $\textbf{Q: Lack of comparison against ........  into consideration label dependency.  } $
>
> $\textbf{Ans:}$ We thank the reviewer for suggesting comparisons with recent state-of-the-art multi-label losses (Ridnik et al., 2021; Cheng & Vasconcelos, 2024). We have updated our experiments accordingly. The results show that CMLL consistently outperforms both methods across all datasets and metrics, demonstrating clear improvements in classification performance and calibration (see table below).
>
> | Method                               | | | ViT  |     ||    | RN50      |    | |
> |------------------------------------------|-------|------|---|-----|-----------|-------------|-----------|-----|------|
> |.   | ACE| MCE| HL| mAP| | ACE| MCE| HL | mAP |
> |ASY (Ridnik et al., 2021)  | 0.2137 | 0.3925 | 0.059 | 0.9417 | | 0.2137   | 0.3125   |  0.0271  | 0.9406 |
> |SPA (Cheng & Vasconcelos, 2024)  | 0.1584 | 0.2935 | 0.1159 | 0.9245 | | 0.1277   | 0.1907   |  0.0264  | 0.9461 |
> |**CMLL (Ours)**  | **0.002**  | **0.0038**  | **0.031**   | **0.9775**  | |  **0.003**  | **0.0052**  | **0.0241**  | **0.9698**  |
>
> $\textbf{ For MS COCO dataset }$
>
>
>
> | Method                               | | | ViT  |     ||    | RN50      |    | |
> |------------------------------------------|-------|------|---|-----|-----------|-------------|-----------|-----|------|
> |.   | ACE| MCE| HL| mAP| | ACE| MCE| HL | mAP |
> |ASY (Ridnik et al., 2021)  | 0.2301  | 0.2885  | 0.1976  | 0.7935  | | 0.2729  | 0.3255  | 0.2248  | 0.8195  |
> |SPA (Cheng & Vasconcelos, 2024)  | 0.2214  | 0.3215  | 0.1910  | 0.7921  | | 0.2538  | 0.3215  | 0.2129  | 0.8049  |
> |**CMLL (Ours)**  | **0.0145**  | **0.0233**  | **0.1936**  | **0.8226**  | | **0.1232**  | **0.2256**  | **0.1741**  | **0.8248**  |
>
>
> $\textbf{ For WIDER-A dataset }$
>
>
>
>
> | Method                               | | | ViT  |     ||    | RN50      |    | |
> |------------------------------------------|-------|------|---|-----|-----------|-------------|-----------|-----|------|
> |.   | ACE| MCE| HL| mAP| | ACE| MCE| HL | mAP |
> |ASY (Ridnik et al., 2021)  | 0.2917  | 0.3411  | 0.1520  | 0.8853  | | 0.2595  | 0.3804  | 0.1239  | 0.8983  |
> |SPA (Cheng & Vasconcelos, 2024)  | 0.3182  | 0.4867  | 0.1248  | 0.9152  | | 0.2807 | 0.4287   | 0.0845  | 0.9158  |
> |**CMLL (Ours)**  | **0.0247**    | **0.0409**    | **0.0761**    | **0.9368**  |  | **0.1603**  | **0.2189**  | **0.0219**  | **0.9251**  |
>
> $\textbf{ For VOC-2012 dataset }$
>
>
>       We thank the reviewer for their thoughtful feedback and for giving us the opportunity to address their concerns. We have provided detailed responses to each point and hope that our results and clarifications satisfactorily resolve the issues raised. In light of these explanations, we respectfully request the reviewer to reconsider their evaluation and, if deemed appropriate, provide a more favourable rating for our paper.

---

### Meta-Review · Area_Chair_rsEa · 2026-01-06

**Summary:**

The paper concerns multi-label classification (MLC). The Authors introduce a new loss function, referred to as *correlated multi-label loss* (CMLL). They claim that this loss function improves calibration and explicitly captures label interdependencies. They try to analyze its theoretical properties. They also provide experimental results on three benchmark datasets.

All initial scores were below the bar and the Reviewers pointed out several critical weaknesses of the paper, such as a limited experimental setup and issues in the theoretical results. As AC I also checked the paper. Based on that I need to admit that the paper has much more flaws that indicated by the Reviewers.

The Authors appear insufficiently familiar with the MLC literature, which has extensively analyzed loss functions and label dependencies (e.g., "On label dependence and loss minimization in multi-label classification", MLJ 2012, "Consistent Multilabel Classification", NeurIPS 2015, "Surrogate regret bounds for generalized classification performance metrics", MLJ 2017, "Consistent algorithms for multi-label classification with macro-at-metrics", ICLR 2024). The Authors would learn the difference between the so-called conditional and unconditional label dependence. Moreover, they would also find out that binary cross-entropy is Fisher consistent along with its regret bound.

The theoretical part given by the Authors is hard to follow, with many typos, and most likely wrong. The introduced loss function is not Fisher consistent as given by the following example. Consider a multi-label classification problem with two labels $y_1$ and $y_2$ and a single feature $x$ taking two values, $a$ and $b$ with the same probability. The conditional distribution of labels for $x=a$ is given by $P(y_1 = 0, y_2 = 0 | x = a) = 1/3$. $P(y_1 = 1, y_2 = 0 | x = a) = 1/3$, and $P(y_1 = 1, y_2 = 1 | x = a) = 1/3$. For $x=b$ we have, $P(y_1 = 0, y_2 = 0 | x = b) = 1/3$, $P(y_1 = 0, y_2 = 1 | x = b) = 1/3$, and $P(y_1 = 1, y_2 = 1 | x = b) = 1/3$. In this case the optimal decisions for the binary cross-entropy (the proper scoring rules) are:
- for label $y_1$: $\hat{y}_1(a) = \frac{2}{3}$, $\hat{y}_1(b) = \frac{1}{3}$
- for label $y_2$: $\hat{y}_2(a) = \frac{1}{3}$, $\hat{y}_2(b) = \frac{2}{3}$

Correlation between the two labels is $1/3$, but correlation between optimal decisions is $-1$. It is clear that prediction scores can be changed such that correlation will change from $-1$ to $1$ (since there are only two feature values, correlation can be either $-1$, $1$, or undefined). Then, depending on the value of $\lambda$, the optimal decisions for the new loss function will no longer be proper scores.

**Reviewer Concerns:**

In general, the initial reviews could be more thorough as the paper has many additional flaws that were not indicated. In the rebuttal the Authors added additional experimental results, but failed in addressing the comments on flaws of their theoretical results.

**Reviewer Scores:**

All Reviewers would lean towards rejecting the paper.

---

### Decision · Program_Chairs · 2026-01-26

Reject